



# Factors influencing the meridional width of the equatorial deep jets

Swantje Bastin[1,2], Martin Claus[1,3], Richard J. Greatbatch[1,3], and Peter Brandt[1,3]

[1]GEOMAR Helmholtz Centre for Ocean Research Kiel, Kiel, Germany
[2]now at Max Planck Institute for Meteorology, Hamburg, Germany
[3]Kiel University, Kiel, Germany

**Correspondence:** Swantje Bastin (swantje.bastin@mpimet.mpg.de, sbastin@posteo.de)

**Abstract.** Equatorial Deep Jets (EDJ) are vertically alternating, stacked zonal currents that flow along the equator in all three ocean basins at intermediate depth. Their structure can be described quite well by the sum of high baroclinic mode equatorial Kelvin and Rossby waves. However, the EDJs' meridional width is larger by a factor of 1.5 than inviscid theory predicts for such waves. Here, we use a set of idealised model configurations representing the Atlantic Ocean to investigate the contributions of different processes to the enhanced EDJ width. Corroborated by the analysis of shipboard velocity sections, we show that instantaneous widening of the EDJ by irreversible mixing processes contributes more to their enhanced time mean width than averaging over meandering of the jets. Most of the widening due to meandering can be attributed to the strength of intraseasonal variability in the jets' depth range, suggesting that the jets are meridionally advected by intraseasonal waves. Only a weak connection to intraseasonal variability is found for the EDJs' instantaneous widening, corroborating and connecting earlier theories that any process dissipating the EDJs' momentum would broaden them, but that intraseasonal variability maintains, not dissipates, the EDJ.

## 1 Introduction

Equatorial Deep Jets (EDJ) are strong zonal currents in the deep equatorial oceans. They take the form of vertically stacked jets that alternately flow eastwards and westwards along the equator, with a vertical scale of a few hundred metres (Luyten and Swallow, 1976; Hayes and Milburn, 1980; Leetmaa and Spain, 1981; Eriksen, 1982; Youngs and Johnson, 2015). In the Atlantic, they are not steady in time but show downward phase propagation on an interannual time scale. Although the EDJs' generation mechanisms are still not entirely clear, it is thought that they are excited by intraseasonal Yanai waves that originate from instabilities in the western boundary currents and/or between the near-surface ocean currents, for example in the form of Tropical Instability Waves (d'Orgeville et al., 2007; Hua et al., 2008; Ascani et al., 2015; Ménesguen et al., 2019). Although Tropical Instability Waves are generated near the surface, they sometimes excite intraseasonal variability that can propagate to depths of a few thousand metres (Tuchen et al., 2018; Körner et al., 2022), where they can provide energy to the EDJ. Similarly, sources of intraseasonal wave energy were identified within the equator-crossing deep western boundary current in



observations and model simulations (Körner et al., 2022). Apart from exciting the EDJ, the intraseasonal Yanai waves also
continuously maintain them, through a mechanism where the Equatorial Deep Jets deform the Yanai waves which leads to a
non-zero net eddy momentum flux reinforcing the EDJ (Greatbatch et al., 2018; Bastin et al., 2020). The EDJ are important
for oceanic tracer transport at intermediate depth, e.g. contributing to the ventilation of the eastern tropical oxygen minimum
zones (Brandt et al., 2012, 2015), and the deficiency of current ocean and climate models in simulating the jets contributes
to the difficulties that the models have with simulating the observed biogeochemical tracer distributions (Getzlaff and Dietze,
2013; Dietze and Loeptien, 2013). Additionally, the EDJ probably influence climate variables in the surface ocean and lower
atmosphere. Brandt et al. (2011) have found variability at the frequency of the Atlantic Equatorial Deep Jets in observed surface
currents, sea surface temperature, surface winds, and rainfall, while Matthießen et al. (2015) showed an impact of EDJ on the
surface flow of the North Equatorial Countercurrent in idealised model simulations.

As for example d'Orgeville et al. (2007), Ascani et al. (2015) and Matthießen et al. (2017) have shown, the Atlantic EDJ are
dynamically very similar to a resonant equatorial basin mode consisting of equatorial Kelvin and Rossby waves, as described
for an idealised inviscid case by Cane and Moore (1981). In these basin modes, the sum of an eastward propagating equatorial
Kelvin wave and its reflection as westward propagating equatorial Rossby waves becomes resonant at a certain period, which
corresponds to the time it takes the Kelvin and the gravest reflected Rossby waves to travel across the basin, and thus depends on
the basin width and the baroclinic mode of the equatorial waves (Cane and Moore, 1981). The EDJ in the Atlantic correspond
to a basin mode of approximately baroclinic mode 17, with a corresponding resonance period of 4.6 years (Claus et al., 2016;
Bastin et al., 2022).

However, the structure of the EDJ shows some features that deviate from the theoretical appearance of such a sum of inviscid
linear equatorial waves with the corresponding vertical scale. One of them is the jets' cross-equatorial width which has been
consistently observed to be larger by a factor of 1.5 than theoretically expected. Muench et al. (1994) found this enhanced
meridional width for the EDJ in the Pacific Ocean from looking at a zonal velocity section between 3°S and 3°N and at
159°W, averaged over 16 months. For the Atlantic EDJ, a widening by the same factor of 1.5 has been shown by Johnson
and Zhang (2003) from an analysis of shipboard CTD profiles, and has later been confirmed by other studies, e.g. Youngs and
Johnson (2015) and Bastin et al. (2022). Muench et al. (1994) suggested that the widening was an artefact caused by time
averaging over EDJ that meander due to meridional advection by intraseasonal waves.

An alternative theory was put forward by Greatbatch et al. (2012). These authors suggested that lateral mixing of momentum
along isopycnals could explain the enhanced width. This is because the zonal flow of the EDJ is close to being in geostrophic
balance and since at the equator, below the Equatorial Undercurrent, diapycnal mixing is known to be particularly weak (Den-
gler and Quadfasel, 2002; Gregg et al., 2003), dissipation of momentum without a corresponding dissipation of the associated
density anomalies must lead to a broadening of the jets, an idea that goes back to Yamagata and Philander (1985). While it
turns out that the meridional flux of momentum in fact acts to maintain, and not dissipate, the jets (Greatbatch et al., 2018;
Bastin et al., 2020), the basic idea put forward by Greatbatch et al. (2012) remains valid. Indeed, Yamagata and Philander
(1985) demonstrated this mechanism in a shallow water model using Rayleigh friction and Newtonian damping to represent
dissipation of momentum and diapycnal mixing, respectively. A broadening of the EDJ could therefore be explained by dissi-





pation of the momentum of the EDJ in the presence of much weaker diapycnal mixing. As noted by Greatbatch et al. (2012),
this process leads to an irreversible loss of momentum whereas the meandering mechanism suggested by Muench et al. (1994)
is a reversible process with no net loss of momentum. Greatbatch et al. (2012) were able to demonstrate their mechanism using
a shallow water model to simulate the EDJ.

Both Muench et al. (1994) and Greatbatch et al. (2012) suggested that the widening of the EDJ about the equator is connected
to the intraseasonal variability in the deep equatorial ocean, although in the theory of Greatbatch et al. (2012) any processes
which lead to enhanced dissipation of momentum could also play a role. Here, we therefore investigate the relation between
the strength of intraseasonal variability and the cross-equatorial width of the EDJ with the help of different idealised model
configurations of the tropical Atlantic Ocean. Additionally, the relative importance of the two suggested mechanisms, i.e. the
reversible meandering through intraseasonal meridional advection versus the irreversible dissipation of momentum, for the
enhanced mean EDJ width is assessed in these models.

Also the amplitude ratio of the Kelvin and the first meridional mode Rossby wave in the EDJ basin mode could play a role
in setting the width of the EDJ, because the zonal velocity signature of an equatorial Kelvin wave has a larger cross-equatorial
width than that of a first meridional mode long Rossby wave of the corresponding vertical mode. Youngs and Johnson (2015)
have shown that the ratio of the Kelvin and Rossby wave amplitudes varies between the three ocean basins: In the Atlantic, the
Rossby wave seems to dominate the EDJ signal whereas in the Indian and Pacific Oceans, the contributions of the two waves
are more similar. So far, the 1.5-fold widening of the EDJ has mostly been estimated in comparison to the expected width of
an inviscid first meridional mode Rossby wave for the corresponding baroclinic mode, not the expected width of the entire
inviscid basin mode (Muench et al., 1994; Johnson and Zhang, 2003; Youngs and Johnson, 2015). Therefore, the effect of the
Kelvin and Rossby wave amplitude ratio for the meridional EDJ width is also investigated here.

The article is structured as follows: In Section 2, the model configurations as well as analysis methods are described. In
Section 3, the results are presented, starting with an overview of the model configurations' ability to simulate EDJ and a
description of the differences in the cross-equatorial width of the EDJ in Section 3.1. This is followed by Section 3.2 where
the amplitude contributions of the equatorial Kelvin and first meridional mode Rossby wave to the EDJ are discussed. The
importance of meandering versus instantaneous widening of the EDJ in setting their time mean cross-equatorial width is
investigated in Section 3.3, as well as the relationship between the meridional EDJ width and the intraseasonal variability in
the models. Finally, a discussion of the results is provided in Section 4.

## 2  Model and methods

### 2.1  Model configurations

The ocean model that has been used for all simulations shown here is the Nucleus for European Modelling of the Ocean
(NEMO, Madec et al., 2017), Version 3.6. The basic model setup is based on the studies by Ascani et al. (2015) and Matthießen
et al. (2015, 2017), who succeeded in simulating EDJ with an idealised model of the tropical Atlantic, although they used the
Parallel Ocean Program (POP), respectively MITgcm ocean models instead of NEMO. All models are ocean-only simulations



for a basin analogous to the tropical Atlantic, but with closed boundaries at 20°S and 20°N. The horizontal resolution is set to $0.25° \times 0.25°$. Following Ascani et al. (2015), the horizontal mixing of tracer and momentum is parameterised using biharmonic diffusion/viscosity with a coefficient of $-2 \cdot 10^{10} \, \mathrm{m^4 \, s^{-1}}$, and the vertical mixing scheme is Richardson number

dependent (Pacanowski and Philander, 1981) with a background diffusivity of $10^{-5} \, \mathrm{m^2 \, s^{-1}}$. The TEOS-10 equation of state is used for all simulations. All model runs are initialised with a horizontally homogeneous density field derived by horizontally and temporally averaging vertical profiles of tropical Atlantic (between 20°S and 20°N) salinity and temperature from the World Ocean Atlas 2018 (WOA18, Locarnini et al., 2019; Zweng et al., 2019). The in-situ temperature and practical salinity from WOA18 have been converted to conservative temperature and absolute salinity with a Python implementation of the

Gibbs Sea Water Library (Firing et al., 2019). At the surface, the temperature and salinity are restored to their initialisation value with a damping time scale of 30 days, to maintain a reasonable stratification of the water column over time.

The model setup from which all the others are derived is called L200-WIND. Its domain is rectangular with a width of 55° to mimic the Atlantic Ocean at the equator, and it has a flat bottom at 5000 m depth. There are 200 model levels, with a vertical resolution of 5 m close to the surface, approximately 20 m in the depth range of the EDJ, and increasing to 50 m close to the

bottom. L200-WIND is forced with zonally and temporally averaged wind stress from the NCEP/NCAR reanalysis (Kalnay et al., 1996; Kistler et al., 2001). It has a free slip boundary condition at the bottom, because Ascani et al. (2015) found that bottom friction reduces the ability of the model to simulate EDJ. In L200-WIND, reasonably realistic EDJ develop in the model, as shown in Bastin et al. (2020) and also visible in Figure 1 where Hovmöller diagrams of the zonal velocity in the centre of the model basin are shown for all model runs used here (continued in Figure 2).

From L200-WIND, a number of other model setups are derived, all of which are listed with their distinguishing features in Table 1. There are two different vertical resolutions, marked by the number after the L in the configuration name. L200 is the fine vertical resolution, and L75 is a coarser one with 75 levels that is one of the commonly used vertical axes by NEMO ocean models, e.g. by the Global Seasonal forecast system of the MetOffice (MacLachlan et al., 2015). Note that there are some configurations that are named L220; these have the same vertical resolution as the L200 configurations but extend to

depths greater than 5000 m and thus have additional layers at these depths because they include realistic bathymetry. Another difference between the configurations is the forcing. There are two types of forcing applied. First, the wind forcing, i.e. zonally and temporally averaged NCEP/NCAR wind stress, is either switched on or off. Second, there is the IMFC (Intraseasonal Momentum Flux Convergence) forcing, which is a tendency added to the zonal momentum equation in the model at every grid point and time step, as described in Bastin et al. (2020). It is the intraseasonal eddy flux convergence from the meridional

advection term in the zonal momentum equation, i.e.

$$-\frac{\partial(\overline{u'v'})}{\partial y} \tag{1}$$

where the prime denotes variability on time scales smaller than 70 days (or intraseasonal) and the overbar means variability on time scales larger than 70 days. This term is thought to be responsible for energy transfer from intraseasonal waves to the EDJ and other slowly varying currents (for details see Greatbatch et al., 2018). For the IMFC model forcing, the term (1) is

diagnosed from the base model configuration L200-WIND, to be applied to other model configurations, as described in Bastin



**Table 1.** Overview of model runs.

| Name | Wind forcing | IMFC forcing | Bathymetry and coastlines |
|---|---|---|---|
| L200-WIND | yes | no | no |
| L200-edjIMFC | no | edj | no |
| L200-2edjIMFC | no | $2 \times$ edj | no |
| L200-fullIMFC | no | full | no |
| L220-bathy-edjIMFC | no | edj | yes |
| L220-bathy-WIND-edjIMFC | yes | edj | yes |
| L220-bathy-fullIMFC | no | full | yes |
| L75-edjIMFC | no | edj | no |
| L75-fullIMFC | no | full | no |

et al. (2020). There are two different flavours of IMFC forcing in the different model experiments: "edj" includes only one Fourier component of the IMFC, namely that with the frequency of the EDJ; and "full" includes the entire IMFC diagnosed from L200-WIND varying on all time scales. "2edj" has the same forcing as "edj" but multiplied by a factor of 2. The last differences between the model setups concern the use of realistic coastlines and bathymetry. All model configurations that do
not have realistic coastlines and bathymetry are rectangular and have a flat bottom like L200-WIND. The model setups that do have realistic coastlines and bathymetry have linear bottom friction instead of a free slip bottom boundary condition like that flat-bottomed setups. Their northern and southern boundaries are still closed at 20°S and 20°N. In the cases where the IMFC forcing diagnosed from the rectangular L200-WIND is applied to setups with realistic coastlines and bathymetry, it is only applied at points that exist in both configurations and set to zero otherwise. Because we chose the rectangular geometry to fit
the width of the Atlantic basin at the equator, this only happens away from the equator and close to the coasts where it is not important for our analysis of the EDJ.

Some combinations of forcing and parameters that we tested did not support EDJ; e.g. bottom friction and/or realistic bathymetry without IMFC forcing; and L75 without IMFC forcing. These are therefore not included here.

## 2.2 Analysis methods

### 2.2.1 Vertical normal mode decomposition

Because the EDJ, unlike most other large-scale flow patterns in the ocean, are characterized by relatively small vertical wavelengths, it is instructive to separate the velocity field into its different vertical scales. This can be done by expressing the flow field's variation in the vertical as a sum of vertical normal modes, to get a sum of linearly independent components of the velocity field, each of which has its unique vertical structure and varies only in the horizontal and time (for details see e.g.
Kundu et al., 2012, Chapter 13.9.).





To obtain the vertical structure functions of the vertical modes, a mean buoyancy frequency profile from the model, averaged along the equator, is used (in case of the configurations with bathymetry only spanning the depth range where there is no bathymetry, i.e. extending to a depth of approximately 3500 m). The zonal velocity field from the model is then projected onto the vertical structure functions. The resulting $u_n$ for each vertical mode $n$ can then be analyzed separately, e.g. by calculating

vertical mode spectra as shown in Figure 1. The separation into vertical normal modes is also used here to vertically filter the zonal velocity by summing only the $u_n$ of modes 15 to 22, to remove variability different from the EDJ.

### 2.2.2 Quantification of meridional EDJ width

The cross-equatorial width of the EDJ has usually been given as the e-folding scale of the meridional profile of zonal velocity amplitude (e.g. Greatbatch et al., 2012). This is continued here. To ensure comparability across all the meridional width

estimates given in this chapter (e.g. also those of a theoretical inviscid Kelvin or Rossby wave), they are all determined by a fit of a Gaussian of the form

$$g(\theta) = a \cdot \exp\left(-\frac{1}{2}\frac{(\theta - \theta_c)^2}{\sigma^2}\right) \tag{2}$$

to the zonal velocity between $1.5°$S and $1.5°$N. Here, $a$ is the amplitude, $\theta$ denotes latitude, $\theta_c$ is the position of the EDJ core (or maximum), and the meridional width of the EDJ is given by $W = \sqrt{2} \cdot \sigma$. The relatively narrow equatorial corridor between

$1.5°$S and $1.5°$N is taken to reduce the influence of off-equatorial maxima in the zonal velocity field, such as associated with Rossby waves. The equatorial radius of deformation for a gravity wave speed of $16.2\,\mathrm{cm\,s^{-1}}$ (corresponding to the EDJ peak baroclinic mode 19 from L200-WIND) is approximately $0.76°$, well covered by the latitude range used for the fit. The width $W$ of an inviscid first meridional mode Rossby wave of this particular vertical mode, determined by the Gaussian fit as described above, is $0.65°$, that of an inviscid Kelvin wave is $1.06°$. The fitting of the EDJ width is done at all longitudes between $25°$W

and $15°$W where the EDJ are strongest, and the width is then averaged over this longitude range.

The zonal velocity is filtered vertically before analysing the cross-equatorial EDJ width, such that only the EDJ variability is included in the analysis. This is done by decomposing the zonal velocity into vertical normal modes (see previous section) and retaining only the contributions of the vertical modes 15 to 22, which cover the EDJ peak (shown in Figures 1 and 2).

A distinction is made between the quantification of the mean EDJ width and the instantaneous EDJ width. The mean EDJ

width is determined by fitting Eq. (2) to the harmonic amplitude field of the vertically filtered zonal velocity varying at the EDJ period which is approximately 4.4 years in the models, whereas the instantaneous EDJ width as well as the EDJ widening due to meandering are determined by fitting Eq. (2) to temporal snapshots of the vertically filtered zonal velocity field. The instantaneous EDJ width is then given as $W = \sqrt{2} \cdot \sigma$. For the EDJ width due to meandering, the resulting distribution of the parameter $\theta_c$, i.e. the shift of the EDJ core away from the equator, is used to produce a distribution of theoretical first

meridional mode Rossby wave zonal velocity amplitude profiles with corresponding meridional shifts. From these, an average profile is calculated and the e-folding scale of this is determined as described above. Previous studies that reported an increased meridional width of the EDJ generally used an inviscid first meridional mode Rossby wave of corresponding vertical scale and frequency as the theoretical comparison, and on this basis calculated a widening of the EDJ by a factor of 1.5 (Johnson and




Zhang, 2003; Youngs and Johnson, 2015). We shall therefore follow that approach here and use the width of an analytic inviscid

first meridional mode Rossby wave as a basis for comparison.

### 2.2.3 Separation of Kelvin and first meridional mode Rossby wave

The contributions of the equatorial Kelvin wave and the first meridional mode Rossby wave to the EDJ are separated by a regression of the models' zonal velocity field onto the theoretical structures of the two waves, where the meridional profile of zonal velocity of the respective wave is specified and the waves' frequency is fixed to the dominant EDJ frequency, which

in the models is approximately $f_{EDJ} = (4.4 \text{ years})^{-1}$. For both waves, a linear combination of a sine and a cosine with the respective frequency and meridional structure is fitted, such that the problem takes the form of a linear regression with four degrees of freedom. The optimisation problem is given by the following term that is minimised using a Python implementation of a least squares linear regression (*scipy.optimize.lsq_linear*, Version 1.6.2):

$$\cdot \|\mathbf{D}_{i,j} \cdot \boldsymbol{b}_j - \boldsymbol{u}_i\|^2 \tag{3}$$

Here, $\boldsymbol{u}$ is the zonal velocity, $i$ a combined multi-index for time and latitude, $\boldsymbol{b}$ is the coefficient vector to be determined with index $j = (1, 2, 3, 4)$, and the design matrix $\mathbf{D}$ is given by:

$$\mathbf{D} = \begin{pmatrix} \cos(\omega \cdot t_i) \cdot u_K(y_i) \\ \sin(\omega \cdot t_i) \cdot u_K(y_i) \\ \cos(\omega \cdot t_i) \cdot u_{R1}(y_i) \\ \sin(\omega \cdot t_i) \cdot u_{R1}(y_i) \end{pmatrix}^T \tag{4}$$

The superscript $T$ denotes the transpose of the matrix. The angular frequency $\omega = 2\pi f_{EDJ}$ is set to that of the EDJ, $t$ is the time and $y$ the northward distance from the equator in m. The meridional profiles of the zonal velocity signatures of the Kelvin

($u_K$) and first meridional mode Rossby ($u_{R1}$) waves are given by the following equations (cf. Gill, 1982, Chapter 11):

$$u_K(y) = u_{0\_K} \cdot \exp(-\frac{\beta y^2}{2c}) \tag{5}$$

$$u_{R1}(y) = u_{0\_R1} \cdot \exp(-\frac{\beta y^2}{2c}) \cdot \left[ \left(c - \frac{c}{3}\right) \cdot 2^{-1} \cdot H_2\left(\sqrt{\frac{\beta}{c}} y\right) - \left(c + \frac{c}{3}\right) \right] \tag{6}$$

for long equatorial Rossby waves with meridional mode number 1. $u_{0\_K}$ and $u_{0\_R1}$ denote constant amplitude values which for

the fit are chosen such that $u_K(y=0) = u_{R1}(y=0) = 1 \text{ m s}^{-1}$, $\beta = 2.3 \cdot 10^{-11} \text{ m}^{-1} \text{ s}^{-1}$ is the change of the Coriolis parameter with latitude, and $H_2$ denotes the second Hermite polynomial. For the gravity wave speed we use a value of $c = 16.2 \text{ cm s}^{-1}$ corresponding to the EDJ peak baroclinic mode 19 from L200-WIND (the changes in the EDJ peak mode gravity wave speed are negligible between the model runs). $u_K$ and $u_{R1}$ are stretched meridionally before the regression, to account for the





different meridional widths of the EDJ in the model configurations. The stretching factor is determined by detecting the mean
latitude of the off-equatorial minima in the EDJ amplitude field, and dividing this latitude by the latitude of the theoretical
Rossby wave zonal velocity profile zero crossing.

The regression is done at all longitudes and depths. From the resulting coefficients $\boldsymbol{b}$, the amplitude $A$ and phase $p$ of the
Kelvin and Rossby wave can then be computed as:

$$A_K = \sqrt{\boldsymbol{b}(1)^2 + \boldsymbol{b}(2)^2} \tag{7}$$

$$p_K = \arctan\left(-\frac{\boldsymbol{b}(2)}{\boldsymbol{b}(1)}\right) \tag{8}$$

$$A_{R1} = \sqrt{\boldsymbol{b}(3)^2 + \boldsymbol{b}(4)^2} \tag{9}$$

$$p_{R1} = \arctan\left(-\frac{\boldsymbol{b}(4)}{\boldsymbol{b}(3)}\right) \tag{10}$$

## 3 Results

### 3.1 EDJ in the different model experiments

In Figures 1 and 2, Hovmöller diagrams of the zonal velocity from the centre of the model basin are shown for all model
configurations listed in Table 1. Spectra of the zonal velocity from the model basin centre are shown as well, calculated
separately for the different vertical normal modes after mode decomposition. For the configurations with realistic coastlines,
the centre of the basin here means halfway along the equator between the coasts.

The EDJ are visible in the Hovmöller diagrams of equatorial zonal velocity as vertically alternating, downward propagating
bands of currents. In the normal mode spectra, they appear as a peak close to the basin mode resonance curve, at a period of
about 4.4 years. The observed period of the real Atlantic EDJ is, with 4.6 years, slightly larger (Bastin et al., 2022).

Across the nine model runs, the cross-equatorial width of the deep jets varies substantially. Shown in Figure 3 are five-day
means of the vertically filtered (modes 15 to 22 to show only the EDJ variability) zonal velocity at 1000 m depth, spaced one
year apart, for L200-WIND and L200-edjIMFC. The EDJ are visible as strong zonal current bands on the equator, changing
direction every few years and propagating from the east towards the west. It is visible that the EDJ in L200-WIND have a larger
meridional scale than those in L200-edjIMFC. Different mechanisms seem to contribute to the enhanced meridional EDJ width
in L200-WIND: there is more meandering about the equator of the EDJ that would lead to a larger meridional scale in the time
mean EDJ signature compared to L200-edjIMFC as suggested by Muench et al. (1994), but the EDJ also seem to have a larger
instantaneous meridional width in L200-WIND than in L200-edjIMFC which could indicate an irreversible widening through
enhanced dissipation of momentum following Greatbatch et al. (2012). The influence of these two different factors on the time
mean meridional width is investigated for all nine model configurations in Section 3.3.

The mean meridional EDJ width for the nine model runs is shown in Figure 4, together with the meridional width of an
inviscid first meridional mode Rossby wave and an inviscid equatorial Kelvin wave with a corresponding vertical structure.
The model runs are sorted by the mean meridional width of the EDJ, from small to large; this order will be kept for the



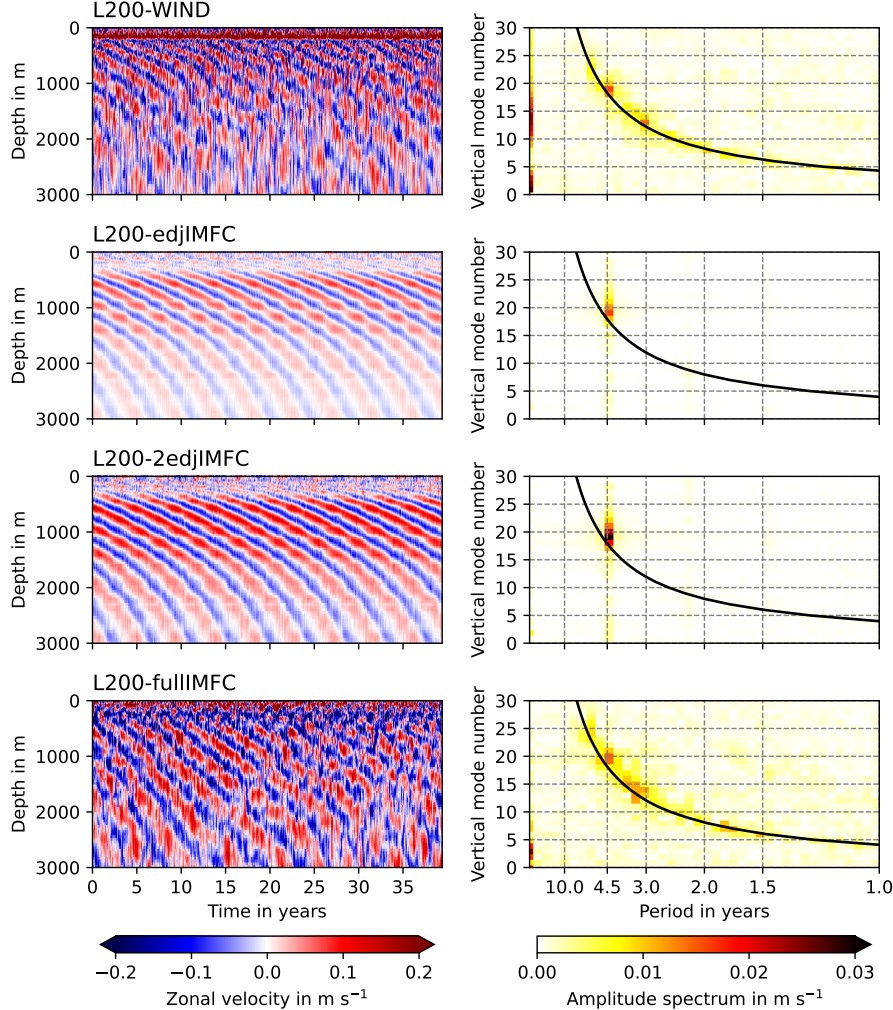

**Figure 1.** Hovmöller diagrams (left panels) and normal mode spectra (right panels) of the zonal velocity in the centre of the model basin, for the model configurations L200-WIND, L200-edjIMFC, L200-2edjIMFC, and L200-fullIMFC. Positive zonal velocity values indicate eastward velocity, negative values mean westward velocity. The solid black line in the right panels shows the resonance frequency for the gravest equatorial basin mode for each vertical normal mode. The spinup is excluded from all model runs. The normal mode structure functions are provided in the supplementary dataset, see data availability statement. (Continued for other model runs in Figure 2.)

following similar figures. Also marked in Figure 4 is the value of the Rossby wave width enhanced by a factor of 1.5. The exact value of the meridional Rossby wave scale is a bit arbitrary, since it depends on the choice of the vertical normal mode of the wave. In fact, the EDJ are composed of multiple normal modes rather than one distinct mode. Here the EDJ peak mode from L200-WIND is chosen (mode 19) for the theoretical inviscid wave widths just to give a visual impression of how much wider the observed EDJ are than the expected inviscid Rossby wave width (the grey dashed line compared to the black dashed





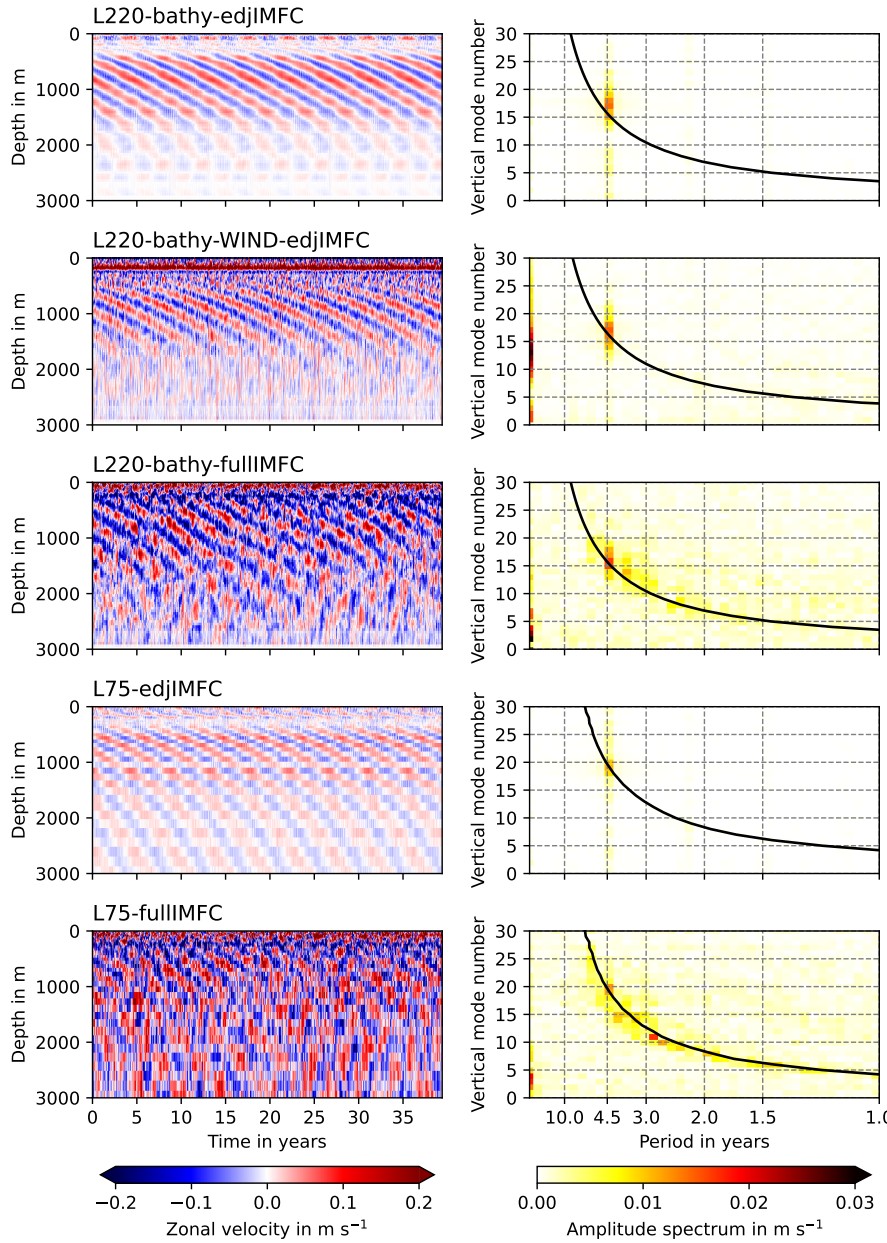

**Figure 2.** As Figure 1, but for the model configurations L220-bathy-edjIMFC, L220-bathy-WIND-edjIMFC, L220-bathy-fullIMFC, L75-edjIMFC, and L75-fullIMFC.

line). The values for the Rossby wave width are a bit smaller here than usually obtained from observations of the Atlantic EDJ: Youngs and Johnson (2015), for example, estimated the Rossby wave width to be 0.73°, consistent with the lower vertical mode number 17 that they found for the Atlantic EDJ peak from observations.



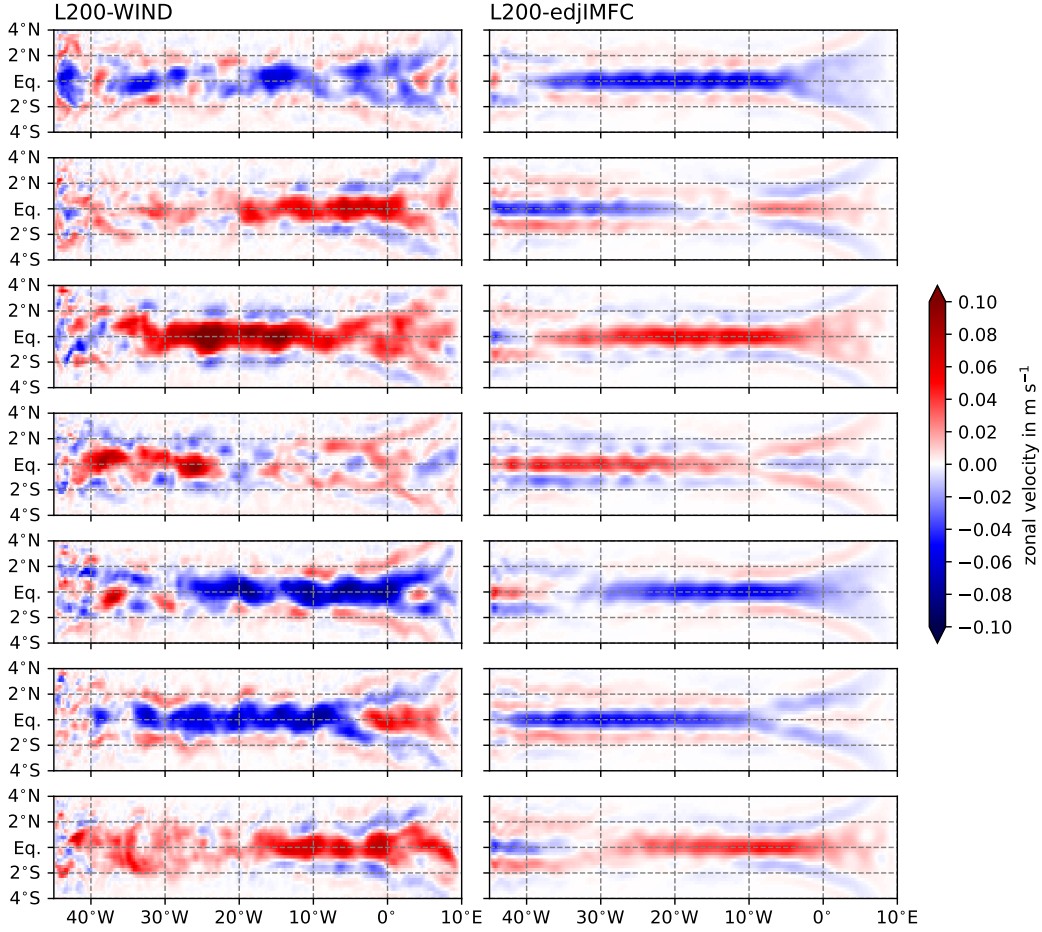

**Figure 3.** Five-daily means of zonal velocity at 1000 m depth in the L200-WIND and L200-edjIMFC model runs. The panels from top to bottom are each one year apart. The zonal velocity has been vertically filtered to contain only the baroclinic modes 15 to 22. Positive zonal velocity values indicate eastward velocity, negative values mean westward velocity.

As mentioned in Section 1 and shown in Figure 4, the Kelvin wave has a larger meridional scale than the first meridional mode Rossby wave. Since the EDJ are composed of the sum of Kelvin and Rossby waves, their amplitude ratio affects the meridional scale of the EDJ. Estimated using the waves' meridional profiles given in Eqs. 5 and 6, the Kelvin wave amplitude would need to be 2.3 times as large as that of the Rossby wave to reach a width that is larger by a factor of 1.5 than the Rossby wave width. This is unlikely to be the main factor responsible for the observed widening of the EDJ, because the Kelvin wave amplitude has been observed to be approximately as large (Indian and Pacific Oceans) or smaller (Atlantic Ocean) than the first meridional mode Rossby wave amplitude Youngs and Johnson (2015). In fact, the contribution of the amplitude ratio seems to be small, because Johnson and Zhang (2003) and Youngs and Johnson (2015) observed the widening by a factor of 1.5 when analysing only the first meridional mode Rossby wave part of the EDJ for all ocean basins. However, the influence of the two





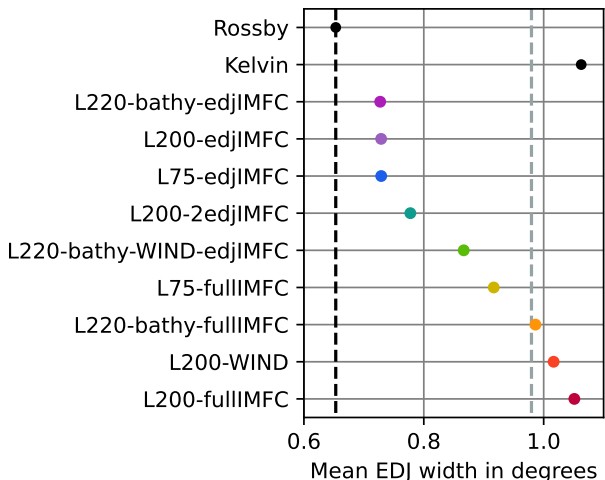

**Figure 4.** Mean cross-equatorial EDJ width in the different model experiments. For the nine model configurations that can simulate EDJ, the e-folding scale of a Gaussian fit to the time mean EDJ amplitude field between 25°W and 15°W is shown. The time mean EDJ amplitude field has been determined as the harmonic amplitude at the EDJ period (4.4 years) of the vertically filtered (modes 15 to 22) zonal velocity field from the models. Also shown are the corresponding meridional widths of an inviscid first meridional mode Rossby wave (black dashed line), 1.5 times the width of the Rossby wave (grey dashed line), and an inviscid Kelvin wave of baroclinic mode 19.

waves' amplitude ratio on the modelled meridional EDJ widths is investigated in Section 3.2, because it might explain part of the differences between the models.

In general, the mean meridional width of the EDJ is larger in those model configurations with wind forcing or fullIMFC
forcing. In contrast to that, models with only edjIMFC forcing, i.e. IMFC forcing varying only at the interannual EDJ frequency, have the narrowest EDJ, although the doubling of the edjIMFC forcing in L200-2edjIMFC leads to a slight EDJ widening compared to L200-edjIMFC. Since variability on all other time scales is much reduced in the model runs forced only at the interannual EDJ frequency, the narrow EDJ in those models could be connected to a lack of variability, e.g. intraseasonal waves. This is investigated in Section 3.3, where also the contributions of meandering, and instantaneous widening, to the time
mean widening of the EDJ are separated.

## 3.2    Contributions of Kelvin and first meridional mode Rossby wave to the EDJ basin mode

In Figure 5, a regression of the vertically filtered (containing only vertical modes 15 to 22) zonal velocity between 4°S and 4°N from L200-WIND on the zonal velocity signature of an analytic equatorial Kelvin and an analytic first meridional mode Rossby wave of vertical mode 19 at the EDJ frequency is shown. The meridional profiles of the two waves have been stretched
meridionally before the regression for every model separately, to account for the different meridional widths of the EDJ. For more details see the methods section. It can be seen in Figure 5 that the Rossby wave has a much larger amplitude at the equator than the Kelvin wave in L200-WIND. This is consistent with observations of the Atlantic EDJ (Youngs and Johnson,





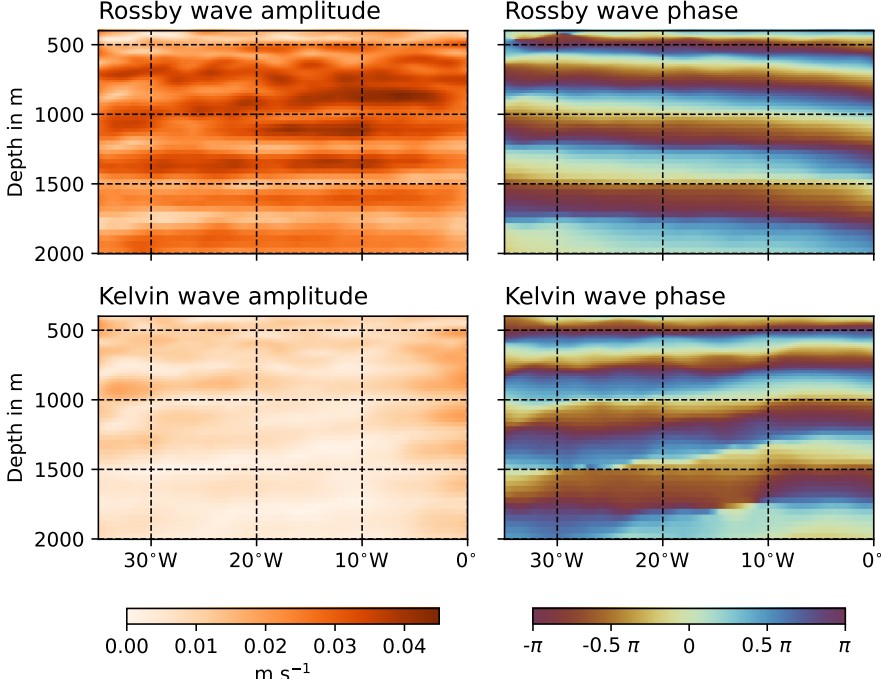

**Figure 5.** Regression of the vertically filtered (modes 15 to 22) zonal velocity from L200-WIND on the zonal velocity signature of an equatorial Kelvin and a first meridional mode Rossby wave of vertical mode 19 and the EDJ frequency. The amplitude shown on the left is the amplitude on the equator.

2015; Bastin et al., 2022). The phase fields of the waves are smooth and correctly show westward propagation for the Rossby wave and eastward propagation for the Kelvin wave, which indicates that the separation of the two wave components by the regression seems to work well. Additionally, the phase difference between the two waves varies approximately linearly with longitude between $-\pi$ at one boundary and $\pi$ at the other boundary for all models (not shown), consistent with the theoretical phase difference of the Kelvin and the first meridional mode Rossby wave in a resonant equatorial basin mode (Cane and Moore, 1981).

To quantify the ratio between the Kelvin and the first meridional mode Rossby wave amplitudes, the amplitude fields from the regression are averaged between 500 and 2000 m depth, and between 25°W and 15°W, where the total EDJ amplitude is strongest and also all other width analyses in this study are performed. The resulting equatorial amplitude values of the two waves are shown for each model in the left panel of Figure 6, together with the amplitude contributions of the Kelvin and first meridional mode Rossby wave to the real Atlantic EDJ basin mode at 1000 m depth as estimated from Argo float data by Bastin et al. (2022). In the centre panel, the amplitude ratio is shown. Again, the model configurations are sorted by the mean meridional width of their EDJ; no systematic relationship between the amplitude ratio and mean EDJ width is visible. In the right panel, the cross-equatorial width of a theoretical basin mode is shown, consisting of an equatorial Kelvin and a first



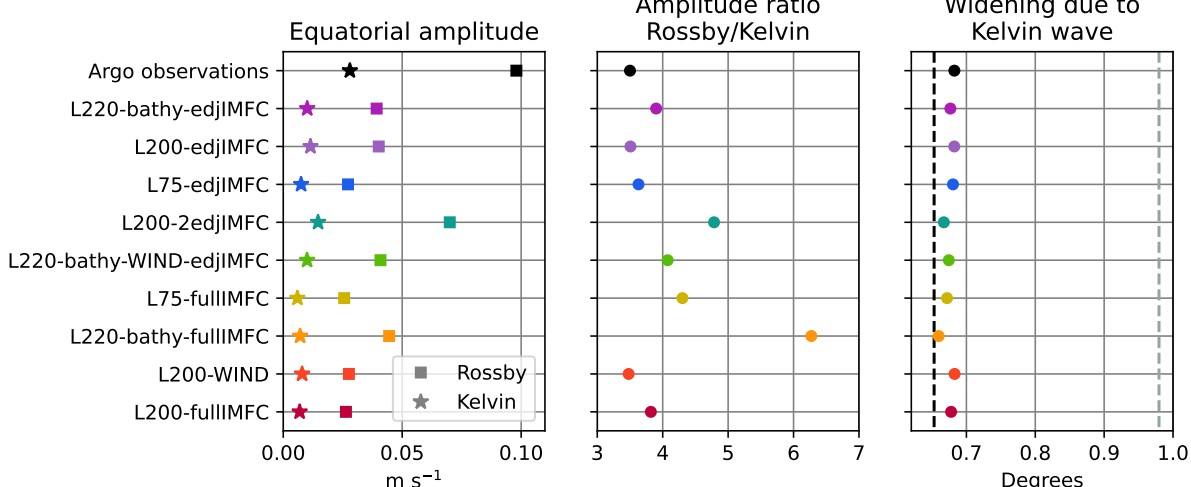

**Figure 6.** Amplitudes of equatorial Kelvin wave and first meridional Rossby wave contributions to the EDJ for each of the nine models with EDJ, as determined by a regression of the waves' zonal velocity signatures on the vertically filtered (modes 15 to 22) zonal velocity field from the models. The models are sorted by their mean EDJ width, from smallest to largest as in Figure 4. Also shown in black are the amplitudes of the waves in the real EDJ, as determined from Argo float data by Bastin et al. (2022). Shown in the right panel by the coloured markers is the cross-equatorial width of a theoretical basin mode with the amplitude ratio from the central panel, and with higher meridional mode Rossby waves neglected. Again, the width (width enhanced by 1.5) of the Rossby wave is marked by the dashed black line (dashed grey line).

meridional mode Rossby wave with the amplitude ratio shown in the centre panel and the phase difference derived by Cane and Moore (1981) but without any higher meridional mode Rossby waves. As expected, a larger relative contribution of the Kelvin wave (smaller ratio in the centre panel) leads to a wider EDJ basin mode, but the effect is very small for the ratios derived 285 from the model solutions and also from Argo data. The differences in the Kelvin wave amplitude compared to the Rossby wave amplitude can thus be rejected as a possible explanation for the differences in the mean meridional EDJ width in the models.

There are a few interesting points to note here that are not related to the cross-equatorial width of the EDJ. In general, the EDJ in the models are much weaker than in the real Atlantic Ocean, as visible in the left panel of Figure 6. One possible reason for this is that the intraseasonal variability exciting and maintaining the EDJ (Ascani et al., 2015; Greatbatch et al., 290 2018; Bastin et al., 2020) is too weak in the model because of the steady or missing wind forcing and thus less instability in the upper ocean currents. However, the amplitude ratio of the Rossby and Kelvin wave is quite realistic in many of the model solutions, although in a few the Rossby wave is even more dominant than in reality. It is intriguing that the models despite their high degree of idealisation correctly simulate the larger Rossby wave amplitude compared to the Kelvin wave amplitude, because this large ratio seems to be a peculiarity of the EDJ in the Atlantic Ocean: Youngs and Johnson (2015) report that 295 in the Indian and Pacific Oceans the Kelvin and first meridional Rossby waves have similar amplitudes. It is not clear where this difference comes from. Possible reasons might include differences in the ocean basins' bathymetry or in the structure





of the coastlines. However, because the amplitude ratio is so close to that of the real Atlantic EDJ in our model runs, both with rectangular geometry and realistic Atlantic geometry, our results suggest that it might rather be the similar amplitudes of Kelvin and Rossby waves in the Indian and Pacific EDJ that are exceptional instead of the Atlantic EDJ characteristics with
the dominant Rossby wave component.

### 3.3    Contributions of EDJ meandering and EDJ instantaneous width, connection to intraseasonal meridional velocity variability

According to the theory of Muench et al. (1994), the EDJ widening could be attributed to time averaging over meandering EDJ, where the meandering is caused by meridional advection by intraseasonal waves. Greatbatch et al. (2012), on the other hand,
proposed that the EDJ are wider than an inviscid basin mode of corresponding vertical scale because of strong dissipation of momentum, perhaps due to small scale velocity fluctuations associated with intraseasonal waves, compared to weak diapycnal mixing of density around the equator. In both cases, the meridional width of the EDJ should thus depend on the spectral power of the intraseasonal meridional velocity variability, although for the theory of Greatbatch et al. (2012) also other processes could play a role that contribute to enhanced dissipation of momentum. More recently, it has been shown that intraseasonal waves
actually lead to a net positive energy influx into the EDJ, thereby maintaining the jets against dissipation rather than weakening them (Greatbatch et al., 2018; Bastin et al., 2020). Therefore, we would expect that the strength of the intraseasonal variability contributes to the meandering of the EDJ, but not so much to the instantaneous widening of the EDJ through dissipation. The strength of the intraseasonal variability at depth shows large differences across the nine model configurations that can simulate EDJ. In Figure 7, spectra of the meridional velocity at the equator and 23°W are shown for moored observations and two
example model configurations, L200-WIND and L200-edjIMFC, which have relatively wide and narrow EDJ, respectively. We use the meridional velocity here to quantify the strength of the intraseasonal variability, because it consists mostly of Yanai waves which only have a meridional velocity component at the equator. It can be seen that the intraseasonal variability at depth is a bit too weak in L200-WIND compared to observations. One reason for this is probably the missing seasonal cycle in the model's wind forcing, because in reality the generation of intraseasonal waves in the tropical Atlantic Ocean is strongest
in boreal summer when the shear instabilities between the surface currents intensify (e.g. von Schuckmann et al., 2008). In L200-WIND, this peak generation of intraseasonal variability is missing because of the steady forcing. Another difference is a shift of the maximum spectral power between 1000 and 2000 m depth towards longer periods of about 50 days in L200-WIND compared to 30-40 days in the moored observations. Nevertheless, L200-WIND shows, as the moored observations, significant spectral power of the intraseasonal meridional velocity variability on the equator, down to depths of at least 3000 m.
In contrast to that, L200-edjIMFC shows very much reduced equatorial meridional velocity variability on all time scales, also in the intraseasonal period range between 30 and 90 days. This is due to the missing wind forcing in this model configuration. Still, there is an intraseasonal peak in the spectrum at a period of about 70 days.

The averaged spectral power of the equatorial intraseasonal meridional velocity variability is shown for each of the nine model configurations in Figure 8 (centre left panel). Again, the models are sorted by their mean meridional EDJ width, which
is also shown in the left panel of the figure for comparison. The relationship between the mean EDJ width and the strength





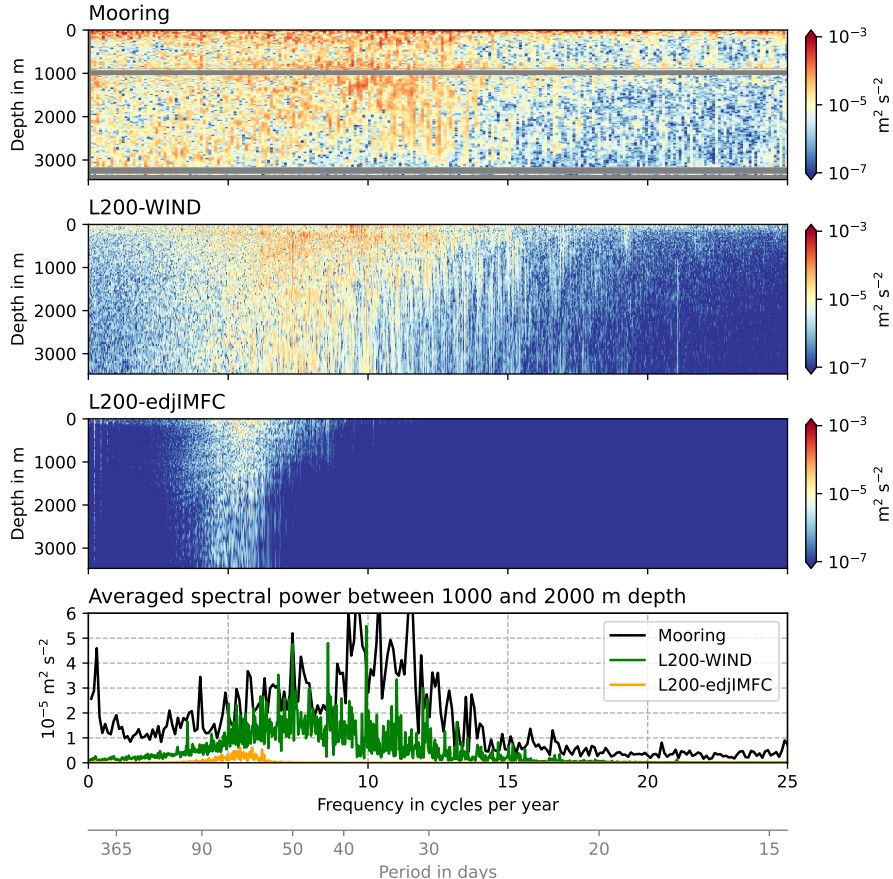

**Figure 7.** Comparison of power spectra of the meridional velocity in moored observations and two model runs, at 0°N, 23°W. The top panel has been updated and modified from Tuchen et al. (2018).

of the intraseasonal variability in the models is not obvious. Therefore, the contributions of meandering and instantaneous widening to the time mean EDJ width are calculated first, followed by the analysis of their relationship with the strength of the intraseasonal variability separately.

To separate the reversible (meandering) and the irreversible (instantaneous widening) part of the EDJ widening in the model, a Gaussian bell curve is fitted to the vertically filtered (modes 15 to 22) zonal velocity at 1000 m depth, at every longitude between 25°W and 15°W and every point in time separately. The mean width due to meandering, as well as the instantaneous width are then estimated from the fit parameters as described in Section 2.2. In Figure 8, the resulting meridional widths for each of the nine model configurations are shown, in the centre right panel for meandering, and in the right panel for instantaneous estimates. Again, the width (width enhanced by 1.5) of the Rossby wave is marked by the dashed black line (dashed grey line), to give an impression of what fraction of the observed mean EDJ widening can be achieved by the process in question. It can be seen that there are differences between the model configurations in how much the EDJ meander. Not surprisingly, the model





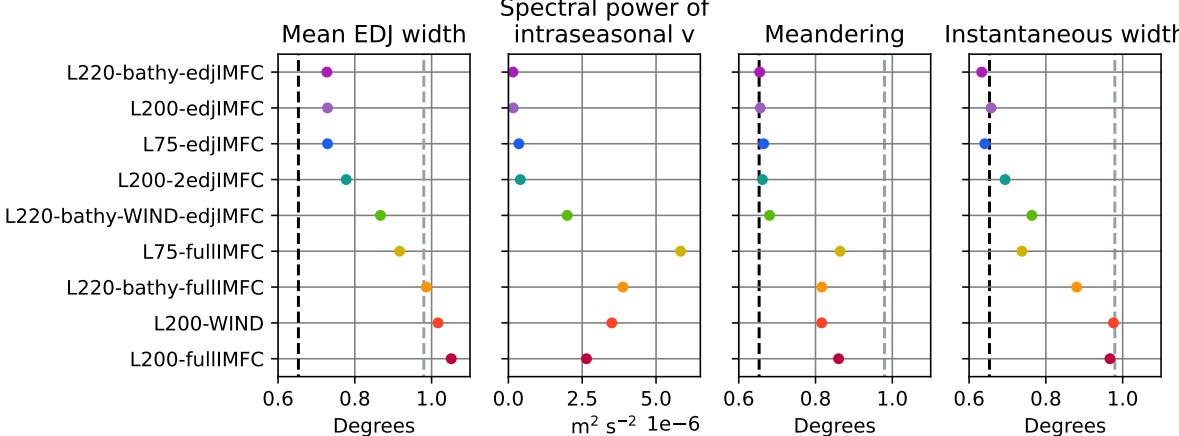

**Figure 8.** Contributions to mean meridional EDJ width by meandering and by instantaneous widening. Left panel: Mean meridional EDJ width, as shown in Figure 4. Centre left panel: Spectral power of the equatorial intraseasonal meridional velocity, averaged over periods of 30 to 90 days, between 400 and 2000 m depth and over all longitudes excluding 7.5° at the western and eastern boundary. Centre right panel: Time mean width of a theoretical Rossby wave meandering with the same distribution of velocity core shifts away from the equator as determined for the model in question. Right panel: Instantaneous meridional width of the EDJ. Again, the width (width enhanced by 1.5) of the Rossby wave is marked by the dashed black line (dashed grey line).

runs with wind forcing or full IMFC forcing, which also have more intraseasonal variability, show more meandering of the EDJ. In the four model configurations with the largest mean EDJ widths, the meandering widens the EDJ in the time mean by about half the observed widening of factor 1.5, in L75-fullIMFC even more. Our model results thus suggest that meandering of

the EDJ does play a role in widening the EDJ meridionally in the time mean. Also the instantaneous width of the EDJ is larger in the model runs with wind forcing or full IMFC forcing, again consistent with the enhanced intraseasonal variability in those models (although here the relationship is not that clear, more details follow in the next paragraph). Except for L75-fullIMFC, the instantaneous widening explains a larger part of the observed EDJ widening than the meandering, with the instantaneous width of the EDJ in L200-WIND and L200-edjIMFC even reaching factor 1.5 compared to the theoretical Rossby wave scale.

As already mentioned, a clear relationship between the strength of the intraseasonal variability and the EDJ meandering can be seen in Figure 8, whereas for the instantaneous widening of the EDJ, the relationship to the strength of the intraseasonal variability in the model is less clear. To investigate and quantify this, linear regressions of the mean width of the EDJ due to the two processes onto the averaged spectral power of the equatorial meridional velocity variability are shown in Figure 9. For both processes, there is a positive correlation between the meridional EDJ width and the strength of the intraseasonal variability. In

the case of the EDJ meandering, the correlation is quite large, and the regression can explain 80% of the variance in mean EDJ width due to meandering. In contrast to that, only 35% of the variance in instantaneous EDJ width can be explained by the regression on the strength of the intraseasonal variability, which is consistent with the more recent finding that intraseasonal waves actually maintain the EDJ against dissipation (Greatbatch et al., 2018; Bastin et al., 2020). Other processes must thus





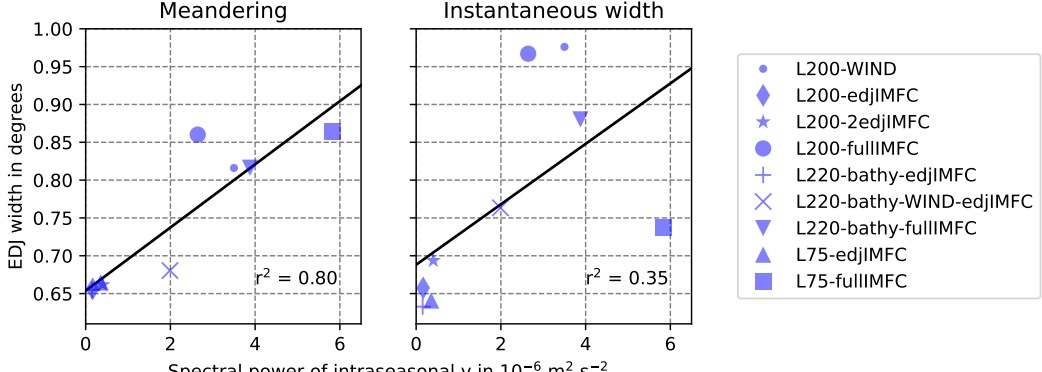

**Figure 9.** Relation between the strength of intraseasonal meridional velocity variability and the enhanced mean meridional width through meandering (left) or instantaneous widening (right) of the EDJ in the models. The spectral power of the equatorial meridional velocity has been averaged over all longitudes excluding 7.5° at the western and eastern boundary, between 400 and 2000 m depth and between periods of 30 and 90 days. Shown in black is a linear regression, with the squared correlation coefficient $r^2$ indicated in the lower right corner.

play the main role in setting the enhanced cross-equatorial instantaneous EDJ width. According to the theory by Greatbatch
et al. (2012), these could be all processes that lead to dissipation of momentum, i.e. cause a negative power input into the EDJ.

## 4   Conclusions and Discussion

In this study, it could be shown that instantaneous widening of the cross-equatorial zonal velocity profile associated with the
EDJ plays a larger role in setting the enhanced time mean meridional width of the EDJ than time averaging over meandering
of the jets. It thus seems that the theory suggested by Greatbatch et al. (2012), which attributes the enhanced meridional EDJ
width to large dissipation of momentum compared to small diapycnal mixing of density, is the main factor for the observed
widening of the EDJ. Nevertheless, also meandering of the EDJ around the equator contributes a non-negligible part to the
meridional widening of the EDJ time mean amplitude field, as suggested by Muench et al. (1994).

However, the results shown here are based only on idealised model simulations of the EDJ. The exact magnitude of the
contributions of both suggested processes in the real ocean cannot be inferred from these model experiments, but it is possible
to gain some insight by looking at instantaneous ship sections of the EDJ. In general, the width of the EDJ has been determined
from observations by looking at time mean sections (e.g. Muench et al., 1994) or by spectral analysis which also gives a time
mean width (e.g. Johnson and Zhang, 2003; Youngs and Johnson, 2015), such that it is not possible to distinguish between the
reversible widening by meandering and irreversible instantaneous widening. By looking at instantaneous (i.e. measured in the
course of a few days) zonal velocity sections, it is possible to assess whether the real EDJ show both an enhanced instantaneous
width and meandering as in the model results presented in this study.





In Figure 10, zonal velocity sections at 23°W are shown which have been measured during several cruises conducted in the frame of the research project *SFB 754, Climate-Biogeochemistry Interactions in the Tropical Ocean* (Krahmann and Mehrtens, 2021; Krahmann et al., 2021). The data have been filtered by decomposing them into vertical normal modes and keeping only modes 14 to 20, approximately corresponding to the EDJ peak, to remove variability different from the EDJ. The normal mode

decomposition has been done using a mean stratification profile from several cruises as described in Claus et al. (2016). To this filtered data, a Gaussian curve has been fitted at all depths between 500 and 2000 m where the maximum velocity exceeds 5 cm s$^{-1}$, and the instantaneous width and width through meandering have been calculated as described in Section 2.2. Also in the cruise data, the instantaneous width of the EDJ is significantly larger than that due to averaging over meandering jets, which corroborates our results from the set of idealised model experiments.

Similar instantaneous zonal velocity profiles of the Atlantic EDJ from different longitudes and times, measured during EQUALANT cruises in 1999 and 2000, are shown e.g. by Gouriou et al. (2001), Bourlès et al. (2003), and Bunge et al. (2006). They also show wider instantaneous EDJ, and only small shifts of the jet cores away from the equator. These observations support the conclusion that instantaneous widening, which can be explained by enhanced dissipation of momentum together with small diapycnal mixing of density (Greatbatch et al., 2012), plays the most important role in setting the enhanced mean

cross-equatorial width of the Atlantic EDJ, whereas averaging over meandering of the jets as suggested by Muench et al. (1994) provides a smaller contribution to the enhanced mean EDJ width.

It has been suggested that the time-mean circulation flanking the EDJ contributes to their enhanced cross-equatorial width by shielding the equator from the effect of Rossby waves that are generated off the equator (Claus et al., 2014). Unfortunately we cannot assess the contribution of that process for the EDJ width in this study, because the strength of the flanking mean

flow in our idealised model simulations is connected to the presence/absence of wind forcing and its effect is thus overlain by that of other wind-driven variability leading to more dissipation. However, Claus et al. (2014) found that the effect of the flanking mean flow on the EDJ width is much smaller than that of enhanced eddy viscosity, which is consistent with our result that momentum dissipation is the most important factor controlling the EDJ width.

Another interesting result from the model experiments shown here is the connection of the meridional EDJ widening to the

strength of intraseasonal variability in the depth range of the EDJ. From the models, it can be concluded that the meandering of the EDJ is very likely largely due to meridional advection of the EDJ by intraseasonal waves, as suggested by Muench et al. (1994). A linear regression of the mean EDJ width due to meandering on the spectral power of the intraseasonal meridional velocity variability in the depth range of the EDJ yields an explained variance of 80%. This is different for the part of the mean EDJ widening that is due to instantaneous widening of the EDJ basin mode. Here, a regression on the spectral power of the

intraseasonal variability also yields a positive correlation, but can explain only 35% of the variance. This in consistent with recent findings, because although Greatbatch et al. (2012) mention in particular intraseasonal meridional velocity fluctuations as a possible source for the enhanced dissipation of EDJ momentum, it has later been shown that intraseasonal waves in fact maintain the EDJ instead of dissipating them (Greatbatch et al., 2018; Bastin et al., 2020). Hence, for the instantaneous widening of the EDJ, other processes and variability on other time scales have to be mainly responsible. For example, the

interaction of the EDJ with the Equatorial Undercurrent (EUC) is a source of momentum dissipation for the jets. Further





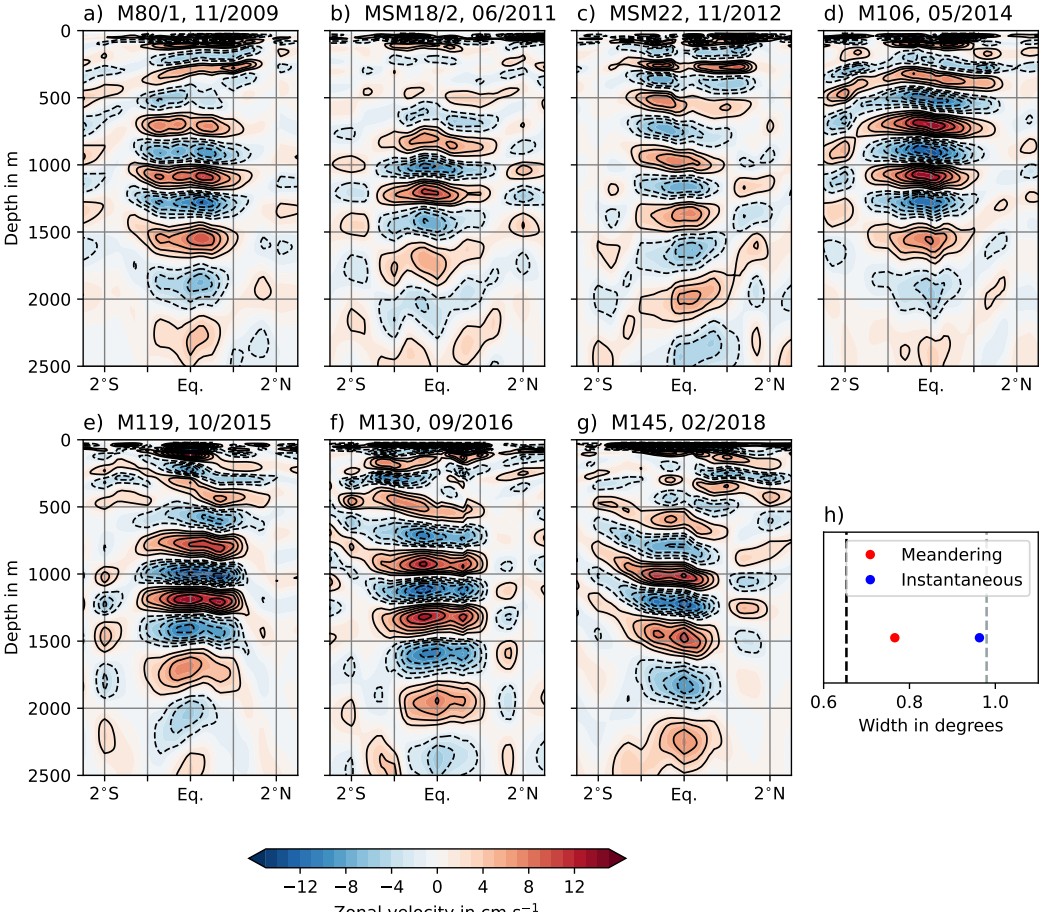

**Figure 10.** Zonal velocity sections measured during seven different cruises along 23°W. Panels a) to g) show the filtered zonal velocity, containing only vertical modes 14 to 20 corresponding to the Atlantic EDJ. Black contours are drawn every 2 cm s$^{-1}$, solid for eastward and dashed for westward velocity. In Panel h), the mean values of instantaneous EDJ width and width due to averaging over meandering jets are shown, again together with the Rossby wave width (dashed black line) and 1.5 times the Rossby wave width (dashed grey line).

research is necessary to identify and quantify the impact of this and other possible sources of momentum dissipation for the EDJ.

*Code and data availability.* Analysis scripts and data necessary to obtain the results presented in this paper can be found online at https://doi.org/10.5281/zenodo.7535589. The cruise data shown in Figure 10 are available at https://doi.pangaea.de/10.1594/PANGAEA.926517
(Krahmann and Mehrtens, 2021; Krahmann et al., 2021).





*Author contributions.* All authors designed the research question and experiments. SB performed the model simulations, carried out the data analysis and created the figures. SB wrote the original draft of the manuscript. MC, RJG and PB contributed to review and editing of the manuscript.

*Competing interests.* The authors declare that they have no conflict of interest.

*Acknowledgements.* Python was used for analysis, Matplotlib (Hunter, 2007) and ScientificColourMaps (Crameri et al., 2020) for visualisation. This work has been funded in part by the Deutsche Forschungsgemeinschaft as part of the *Sonderforschungsbereich 754 – Climate-Biogeochemistry Interactions in the Tropical Ocean* and through several research cruises with RV Meteor and RV Maria S. Merian.



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
