# Peer review of "Factors influencing the meridional width of the equatorial deep jets"

_EGUsphere, 2023_

## Referee Comment (RC1)

**Review of Bastin et al 2023 for Ocean Sciences**

**General comments**

This paper investigates different potential forcing mechanisms for the observed widening (compared to theoretical equatorial waves) of the equatorial deep jets using a range of idealized models. This paper is generally well written with clean figures, but often assuming too much background information from the readers, and the specific messaging of the mechanisms for widening could be cleaned up. Overall, the paper provided a theoretical step forward to understand the puzzling phenomena of Equatorial Deep Jets, and thus suitably significant for publication.

**Specific comments**

1. Abstract. The abstract could be stronger if it ended with implications for this work on our understanding of the deep jets and highlighted the significance of the work.

2. L26-33. This motivation is good, and it might read better if it were placed before the explanation of the potential mechanisms driving the deep jets.

3. Introduction. By the end of this section, I was left wondering what are the general gaps in the fields. Highlighting the gaps progressively as they become narrower to the questions you are addressing would be helpful for background information.

4. Section 2.1. A figure might be helpful to show the model domain and orient the reader. In addiction, the acronyms/handles that you use to explain the model simulations are a bit hard to follow for your analysis. It may be helpful to use descriptors for the model runs when you discuss them in the results and conclusions. In general, if you could use words to describe intraseasonal momentum flux convergence instead of exclusively the acronym may also be helpful.

5. L222-230. This paragraph is confusing because it is hard to follow with the experiment nomenclature, see above. It also contains a bit of discussion that would seem to fit better later in the paper. This paragraph may be better served as a time to walk the reader through Figure 4, as suggested by your topic sentence.

6. L254-260. This paragraph could be moved forward to walk through figure 4, see the previous comment. It is also a bit heavy in jargon, but is a very important result!

7. Figure 5. I am not sure this figure is effective. I am unclear about how it adds to your story. Consider removing it or combining it with figure 3? It seems like its purpose is to show that your model describes deep jets effectively, which would need to be moved forward in the paper.

8. L287-300. This paragraph is really important, but it seems like model validation. Maybe the section 3.2 moves forward as a model validation?

9. Section 3.3. This section is a bit hard to follow in terms of keeping the different mechanisms separate, starting with a very long first paragraph. Is there a way to subsection it out to investigate each potential mechanism separately.

10. Section 4. I am still confused about what the instantaneous deep jet width versus the meandering time mean width. For figure 10h, I expected the meandering width to be larger because it would include the instantaneous component and variability from the meandering, but it is not. Maybe you can clarify in the discussion?

**Technical comments**

1. L6. Instantaneous widening is a bit confusing a term for non-specialists, consider

2. L7-8. This sentence was confusing because in the previous one you said that the meandering wasn't important, and now you are explaining more about it. Maybe in the previous sentence, you can indicate the relative importance of the mechanisms (like about 1/3) instead of implying that the meandering isn't important.

3. L34. This sentence is awkward, consider "d'Orgeville et al. (2007), Ascani et al. (2015) and Matthieen et al. (2017) have shown the"

4. L51-53. This sentence was a bit hard to follow. Consider separating it into two.

5. L141. It may be a personal preference, but it can be awkward to start sentences, especially paragraphs with because.

6. L176-180. I would switch the order of these sentences and instead say we do this and then justify rather than the other way around.

7. Figure 1. How long is the spin up time?

8. L249. Youngs and Johnson (2015) → (Youngs and Johnson 2015)

9. L364-367. Maybe you could quantify the proportions of each mechanism, like 1/10th or 1/3rd to give readers intuition for these processes.

---

## Author Response (AR1)

(The original reviewer comments are in black, our point-by-point replies are in blue. The line numbers in our replies refer to the lines in the diff.pdf showing the changes to the last version.)

**Reply to Reviewer 1**

General comments

This paper investigates different potential forcing mechanisms for the observed widening (compared to theoretical equatorial waves) of the equatorial deep jets using a range of idealized models. This paper is generally well written with clean figures, but often assuming too much background information from the readers, and the specific messaging of the mechanisms for widening could be cleaned up. Overall, the paper provided a theoretical step forward to understand the puzzling phenomena of Equatorial Deep Jets, and thus suitably significant for publication.

Thank you very much for your time and effort, and for the constructive review which helps us to improve the manuscript!

Specific comments

1.  Abstract. The abstract could be stronger if it ended with implications for this work on our understanding of the deep jets and highlighted the significance of the work.

Thanks for the suggestion. We tried to strengthen the abstract by adding a point on the implications/ significance of the work (ll. 11-13).

2.  L26-33. This motivation is good, and it might read better if it were placed before the explanation of the potential mechanisms driving the deep jets.

We rearranged the text as suggested (ll. 18-27, 37-44).

3.  Introduction. By the end of this section, I was left wondering what are the general gaps in the fields. Highlighting the gaps progressively as they become narrower to the questions you are addressing would be helpful for background information.

We added some text on the general gaps in EDJ research (e.g. concerning their generation mechanisms) in the first half of the introduction (ll. 25-27, 59-60).

The specific gaps that we address in the study were already described:

-   Which of the suggested processes by Muench et al. (1994), and Greatbatch et al. (2012) are responsible for the widening of the EDJ?

- How much does the amplitude ratio between Kelvin and Rossby wave, on which there are varying results for the different ocean basins, contribute to the observed EDJ widening?

However, we rewrote the text to identify these gaps more clearly (ll. 85ff, 95-96).

4. Section 2.1. A figure might be helpful to show the model domain and orient the reader. In addition, the acronyms/handles that you use to explain the model simulations are a bit hard to follow for your analysis. It may be helpful to use descriptors for the model runs when you discuss them in the results and conclusions. In general, if you could use words to describe intraseasonal momentum flux convergence instead of exclusively the acronym may also be helpful.

We added a figure showing the model domain (new Figure 1). For the sake of unambiguity, we kept the model run acronyms/handles, but tried to additionally use more descriptive explanations in the text whenever we write about one of the model runs (e.g ll. 136, 140, 260-261). We used the words for intraseasonal momentum flux convergence more often instead of abbreviating it (e.g. l. 275, compare to l. 300 because the text was also rearranged, l. 395).

5. L222-230. This paragraph is confusing because it is hard to follow with the experiment nomenclature, see above. It also contains a bit of discussion that would seem to fit better later in the paper. This paragraph may be better served as a time to walk the reader through Figure 4, as suggested by your topic sentence.

We rearranged this paragraph a bit, as suggested. E.g. kept only the parts that are strictly about the description of Figure 3 in this place, and moved the topic sentence together with the rest of the paragraph towards Figure 4 (ll. 257-280). We will also take care to make the experiment nomenclature more clear (see above).

6. L254-260. This paragraph could be moved forward to walk through figure 4, see the previous comment. It is also a bit heavy in jargon, but is a very important result!

We moved the paragraph as suggested. We also rewrote it to make it more clear, and tried to use less jargon (ll. 273-280).

7. Figure 5. I am not sure this figure is effective. I am unclear about how it adds to your story. Consider removing it or combining it with figure 3? It seems like its purpose is to show that your model describes deep jets effectively, which would need to be moved forward in the paper.

The purpose of Figure 5 is to show that we can effectively separate the Kelvin and Rossby wave components of the EDJ, rather than showing that the model simulates deep jets effectively. So we would argue that it fits better in the section about the contributions of the Kelvin and Rossby waves, where it is now.

8. L287-300. This paragraph is really important, but it seems like model validation. Maybe the section 3.2 moves forward as a model validation?

We would argue that this paragraph is not model validation, but a description of interesting results from the Kelvin and Rossby wave amplitude analysis which the section deals with. However, since the results that are described are not directly related to the width issue that is the main story of the article, we put them to the end of the section. We would like to keep them there.

9. Section 3.3. This section is a bit hard to follow in terms of keeping the different mechanisms separate, starting with a very long first paragraph. Is there a way to subsection it out to investigate each potential mechanism separately.

We rearranged this to make it easier to read, e.g. by separating the text into more paragraphs as suggested, and by rewriting/adding more explanation so that it becomes easier to follow (ll. 348-364).

10. Section 4. I am still confused about what the instantaneous deep jet width versus the meandering time mean width. For figure 10h, I expected the meandering width to be larger because it would include the instantaneous component and variability from the meandering, but it is not. Maybe you can clarify in the discussion?

The meandering width should not contain the instantaneous component, so it does not need to be larger than the latter. We did the analysis in such a way that the two processes should be separated in the results.

The instantaneous width is basically the e-folding scale of a Gaussian fit to the meridional profile of the EDJ. Since we allow the profile to be displaced to the north or south and do the fit at every longitude and time separately, meandering of the EDJ should not influence this width estimation.

To get the part of the EDJ widening that would be due to „time averaging over meandering jets", we also do Gaussian fits at every longitude and time as described before, but in this case we only use the north/south displacement information. This gives us a distribution of displacement values, which we use to generate a number of meridional profiles that have the detected displacement of the EDJ, but the e-folding width of a theoretical Rossby wave. Over these, we then average and thus get a width value due to meandering. In this there should be no influence from the instantaneous width of the EDJ.

We hope this is clearer now, and rewrote the text in the methods section so that it is easier to understand in the article as well (ll. 205-208). (For clarity, we have kept the term „instantaneous widening" in this reply, but we changed it in the manuscript to something like „widening by dissipation or momentum loss", see comment below.)

Technical comments
1. L6. Instantaneous widening is a bit confusing a term for non-specialists, consider

Thanks for the suggestion. We changed the term „instantaneous widening" throughout the manuscript to something more descriptive and easily understandable, e.g. „widening by dissipation or momentum loss".

2. L7-8. This sentence was confusing because in the previous one you said that the meandering wasn't important, and now you are explaining more about it. Maybe in the previous sentence, you can indicate the relative importance of the mechanisms (like about 1/3) instead of implying that the meandering isn't important.

We do not agree that the previous sentence implies that the meandering is not important. It already indicates the relative importance of the mechanisms, saying that „instantaneous widening of the EDJ […] contributes more to their enhanced time mean width than averaging over meandering of the jets."

3. L34. This sentence is awkward, consider "d'Orgeville et al. (2007), Ascani et al. (2015) and Matthieen et al. (2017) have shown the"

We changed this as suggested (l. 45).

4. L51-53. This sentence was a bit hard to follow. Consider separating it into two.

We separated this as suggested to make it easier to read (ll. 64-67).

5. L141. It may be a personal preference, but it can be awkward to start sentences, especially paragraphs with because.

We rearranged this as suggested (ll. 171-172).

6. L176-180. I would switch the order of these sentences and instead say we do this and then justify rather than the other way around.

We rearranged as suggested (ll. 210-214).

7. Figure 1. How long is the spin up time?

The spinup is 50 years for the model runs without IMFC forcing and 10 years for the runs with IMFC forcing. (It is shorter for the latter because there the EDJ develop faster. If they have to develop freely, they take longer.) We added the information to the article (caption of Figure 2, previously Figure 1).

8. L249. Youngs and Johnson (2015) → (Youngs and Johnson 2015)

We corrected this (l. 294).

9.   L364-367. Maybe you could quantify the proportions of each mechanism, like 1/10th or 1/3rd to give readers intuition for these processes.

We added a sentence quantifying the proportions as suggested. However, we suggest to do it later for the cruise observations instead of for the idealised model simulations (ll. 445-447).

**Reply to Reviewer 2**

*Review 'Factors influencing the meridional width of the equatorial deep jets' (Claire Menesguen)*

The study deals with the issue concerning Equatorial Deep Jets meridional width. Even if well explained by the superposition of a Kelvin and a Rossby waves, they are observed to be wider in the meridional direction. Several hypothesis are tested by the mean of idealized simulations.

It is a real pleasure to read such a well written article for a first review. It is very easy to follow the logic of the scientific work. All informations that are needed to the understanding are well organized in the paper, and easy to find back if necessary.

That is why I recommend only very minor revisions.

Thank you very much, Claire, for the interesting suggestions and the constructive review which helps us to improve the manuscript!

One concern I have is to have an idea of error bars in figures where simple markers are used for each experiment (figs 4, 6, 8, 9). I guess, for example, the more variability is in play, the more uncertainty it will create. But it would be worth to quantify that and add error bars.

Thanks for the suggestion. We added a quantification of uncertainty to our analysis and added error bars to the figures. (Since the numbering changed, now see Figures 5, 7, 9, 10 and associated captions)

I wonder if inertial instability can't be at play to explain part of the widening of zonal jets. Indeed, few years ago, Menesguen et al. 2009b have noticed that westward jets (combined with EEJ) could be the place of Inertial instability (Menesguen et al. 2009b : « Intermittent layering in the Atlantic equatorial deep jets »). It could go in the sense of enhanced momentum dissipation within jets, as mentioned by Greatbatch et al. 2012.

We added a discussion on this and mentioned the possibility (ll. 479-483). However, we have some doubts that this mechanism plays a large role because inertial instability would involve some vertical overturning, i.e. mixing of density in the vertical. It has been shown though that diapycnal

mixing is very weak at the equator below the EUC (Dengler and Quadfasel, 2002; Gregg at al., 2003).

One thing I still don't totally get is how a constant wind, zonally averaged, could give enough variability at depth to produce EDJ. Could you briefly describe the spin up? What are the first processus to take place? and how long does the simulation takes to spin up?

The variability at depth comes from instabilities in the sheared upper ocean currents. Because the wind is only zonally averaged, but still varies meridionally, the upper ocean currents are relatively well reproduced and create shear between them as in the real ocean. Thus, also instability waves are generated and also propagate to depth as observed (e.g. Tuchen et al., 2018, Körner et al., 2022). However, the seasonal cycle that is observed in the real ocean is missing in our model because of the steady forcing, and the level of variability at depth is in general a bit lower than in observations. It is still enough to produce EDJ of reasonable amplitude in our idealised case.

The spinup is 50 years long in the model runs without IMFC forcing, because the EDJ take quite long if they develop freely through wave instabilities/interactions. With IMFC forcing, they develop faster and the spinup only has to be 10 years long until their amplitude is stable.

We added these explanations in the article (ll. 133-136, caption of Figure 2).

The next question is, I guess, related to my misunderstanding of the spin up.

Why does a realistic bathymetry, without IMFC forcing, fail to produce EDJ? (With and without wind forcing). I would have expected that IMFC forcing is a part of what happens with the wind forcing. Then, wind forcing should more probably produce EDJ than IMFC forcing, which is not the case.

Could it be the problem in Ascani et al. 2015 simulations with realistic coastlines?

Instead of adding this specific forcing -that comes from a simulation with the simple wind forcing and no dissipation at boundaries-, would it produce EDJ with wind forcing with twice its amplitude?

Why the case with realistic bathymetry and wind forcing does not produce EDJ, we unfortunately still do not understand. It is definitely a similar problem as in Ascani et al. (2015). Even the addition of bottom friction in the flat-bottomed, rectangular case prohibits the formation of EDJ in the model. Somehow the momentum balance for the generation and maintenance of the EDJ is not entirely correct in the model, probably because some processes are missing. That seems to make them overly sensitive to additional sources of dissipation, e.g. bottom friction.

As you say, the IMFC forcing is a part of what happens with the wind forcing. However, it is important to note that the IMFC forcing is the result of the interaction of wind-generated intraseasonal waves **and** the EDJ themselves. This means that if no EDJ develop in the model, there is also no intraseasonal momentum flux convergence at the EDJ frequency. This is why we have to take it from a model run that has EDJ by itself.

Using wind forcing with twice the amplitude is an interesting suggestion. We had not tried this, but have now made an additional model run with doubled wind forcing as you proposed. You can see

the zonal velocity as a function of depth and model year in the figure above. There are some hints of EDJ-like structures evolving around years 20-25, but they disappear and instead there are different alternating currents that show upward instead of downward phase propagation. These are so different in structure from the Atlantic EDJ, though, that we cannot use them for our investigation of the enhanced EDJ width.

L.122: why 70 days? Why not 90 days? Are results sensible to this cut-off period?

We took a cutoff period of 70 days to be consistent with Greatbatch et al. (2018), but we calculated the IMFC with a cutoff period of 90 days as well and the results were not sensitive to this.

Fig7: it could be easier to read the Period axe if this one was on top of the first subpanel, and if the ticks were reported on top of to the other subpanels.

We changed this as suggested (now Figure 8).